# Sustainable Fabrication of Organic Solvent Nanofiltration Membranes

**DOI:** 10.3390/membranes11010019

**Published:** 2020-12-28

**Authors:** Hai Yen Nguyen Thi, Bao Tran Duy Nguyen, Jeong F. Kim

**Affiliations:** 1Department of Energy and Chemical Engineering, Incheon National University, Incheon 22012, Korea; haiyen0107@inu.ac.kr (H.Y.N.T.); bao.nguyen@inu.ac.kr (B.T.D.N.); 2Innovation Center for Chemical Engineering, Incheon National University, Incheon 22012, Korea

**Keywords:** sustainability, organic solvent nanofiltration, green solvents, environmental-friendly polymers, bio-based polymers

## Abstract

Organic solvent nanofiltration (OSN) has been considered as one of the key technologies to improve the sustainability of separation processes. Recently, apart from enhancing the membrane performance, greener fabricate on of OSN membranes has been set as a strategic objective. Considerable efforts have been made aiming to improve the sustainability in membrane fabrication, such as replacing membrane materials with biodegradable alternatives, substituting toxic solvents with greener solvents, and minimizing waste generation with material recycling. In addition, new promising fabrication and post-modification methods of solvent-stable membranes have been developed exploiting the concept of interpenetrating polymer networks, spray coating, and facile interfacial polymerization. This review compiles the recent progress and advances for sustainable fabrication in the field of polymeric OSN membranes.

## 1. Introduction

With escalating environmental concerns, the concept of sustainability has become more important. The environmental regulations are getting even more stringent, and industries are widely implementing membrane technology to improve their process sustainability. Among different types of membrane technologies, organic solvent nanofiltration (OSN) is a membrane process with capabilities to discriminate nanometer-sized molecules in organic solvents [1]. Different from ultrafiltration and reverse osmosis, nanofiltration membranes exhibit pore size distributions of 0.5–2 nm [2]. OSN has been recognized as a sustainable separation platform with low energy consumption [3] and, also, as a platform to achieve process intensification via reaction-separation convergence [4].

In order to perform separation in organic media, OSN membranes are required to resist harsh media such as aggressive solvents; simultaneously, OSN membranes must be stable over a wide range of pH, including organic acids and bases [4]. Hence, most research efforts have been dedicated on improving the chemical stability of polymeric membranes. However, modifications such as crosslinking employ toxic reagents and chemicals that generate considerable amount of organic wastes. Ironically, although OSN technology has developed to improve the process sustainability, the fabrication of OSN membrane itself has been far from sustainable [3].

As discussed in the work by Figoli et al. [5], to be “sustainable”, a membrane process must not implicate the use of dangerous chemicals in the membrane production process itself. Moreover, a sustainable development can only be realized when it satisfies the current needs without undermining the ability of next generations to meet their own needs. Ideally, a process should comply with green principles [6], such as effective waste prevention, better atom economy, replacement of hazardous chemicals with safer alternatives, and higher energy efficiency.

Recently, there has been an encouraging progress with respect to sustainability in the OSN membrane fabrication. There are many techniques to fabricate and to modify membranes, such as phase inversion, interfacial polymerization, sintering, irradiation and track-etching, dip coating, and plasma polymerization [7]. Among the reported methods, the phase inversion technique has been the most employed one due to its versatile feature [8] to control the membrane morphology, pore connectivity, and pore size distribution [9].

There are different types of phase inversion methods, as illustrated in Figure 1. The principle behind all the methods is essentially the same. Among these illustrated methods, nonsolvent induced phase separation (NIPS) is the most widely employed one for making commercial membranes due to its versatility, closely followed by the thermally induced phase separation (TIPS) technique.

In NIPS, the homogeneous dope solution is first prepared by dissolving a polymer(s) in a suitable solvent(s). Additives could be supplemented to enhance the desired properties of the membrane. The prepared solution is then cast onto a substrate to make a thin film, which is subsequently immersed in a nonsolvent bath (thermodynamically unstable environment—mostly water) for coagulation. The phase separation and solidification proceeds spontaneously [13]. Notably, the NIPS process inevitably generates a significant amount of solvent-contaminated wastewater, which must be treated on-site or off-site before discharging into the environment [13].

In comparison, the TIPS dope solution is prepared at an elevated temperature and the membrane is formed by cooling the cast solution below its solidification point. For vapor/evaporation (VIPS/EIPS), the cast solution comes in contact with a humid environment, where vapor absorbs onto the film with evaporation of the dope solvent in tandem [13].

Depending on the target membrane properties, post-modification or post-treatment steps can be applied using techniques such as crosslinking, coating, and interfacial polymerization. Important components and parameters for each step of the phase inversion method are shown in Figure 2. The sustainability of each factor must be considered and improved.

Two key components in the phase inversion method are the polymer and solvent. Common types of polymer materials and solvents used to fabricate membranes were compiled by Lee et al. [14], as shown in Figure 3. Within the membrane technology, current OSN membranes reported in the literature are mostly petroleum-derived polyimide (PI), polybenzimidazole (PBI), and poly(ether ether ketone) (PEEK) (please refer to the list of abbreviations after the Conclusions Section). On the other hand, common solvents used to fabricate membranes now are mostly toxic polar aprotic solvents such as dimethylformamide (DMF), dimethylacetamide (DMAc), and N-methylpyrrolidone (NMP). To maintain the sustainable development of membrane technology, it is imperative to replace these toxic materials with environmental-friendly alternatives, as the use of these solvents will be restricted after May 2020 in Europe [15]. In fact, this has been an active research topic around the globe for this very reason.

In the past two decades, many promising advancements have been made by researchers to realize the concept of sustainability in membrane technology. In this review, the reported greener alternatives for membrane polymers and solvents were compiled and compared. Additionally, the recent progress in sustainable membrane fabrication methods were also assessed.

## 2. Sustainable Membrane Materials

There has been an active search to substitute petro-derived conventional polymers with natural alternatives such as bio-based polymers. The first generation of bio-based polymers employed edible crops as raw materials; however, due to the competition for land used in food production, the focus has gradually shifted to nonedible crops [16]. Some natural products that could be considered as bio-based polymers are collagen, chitosan, and others that could be synthesized by bio-routes [17].

Recently, Peinemann et al. [18] reported an interesting work to utilize a sodium alginate (NaAlg) polymer to fabricate composite membranes. This NaAlg polysaccharide originated from a seaweed cell wall that contained 30% to 60% alginic acid, which can be converted to a water-soluble salt. The modified polysaccharide is an unbranched binary copolymer of 1-4-linked β-D-mannuronic acid (M) and α-L-guluronic acid (G). The alginate polymer has numerous advantages, including the ability to form uniform and transparent films at ambient temperature, their abundance, biodegradability, and characteristic liquid-gel behavior in aqueous solutions.

The proposed work employed a biodegradable alginate as a membrane polymer, water as the solvent, and calcium chloride as the ionic crosslinking agent. The authors applied the technique called RIPS (new type of phase inversion method) to prepare sodium alginate composite membranes [19], as schematized in Figure 4. The prepared dope solution was cast onto different support materials (polyacrylonitrile (PAN), cellulose, nonwoven polyester, and alumina discs). The cast solution was then left to dry at room temperature instead of immersion in a nonsolvent media. In step (II), the aqueous calcium chloride solution was used as the crosslinking solution, further enhancing the structural stability of the membranes and sustainability of the membrane fabrication. Depending on the desired membrane property, the dried film can be immersed into a nonsolvent bath (step III in Figure 4).

The fabricated membranes were surprisingly stable in DMF, dimethyl sulfoxide (DMSO), and NMP [18]. The membranes also exhibited a competitive and stable OSN performance, as compiled in Table 1. In addition, NaAlg membranes showed a good performance for CO_2_/N_2_ gas separation, and it can be a very economical polymer compared to commercial membranes like Pebax^®^ and Matrimid^®^ 5218 [16].

Polylactic acid, or polylactide (PLA), is a polyester derived from renewable resources such as starch, roots of tapioca, or sugarcane. In 2010, PLA was the second-most important bioplastic in the world, following the starch-based polymer in terms of consumption volume [20]. Its name of “polylactic acid” is not in-line with IUPAC standard nomenclature, because PLA is not a polyacid (polyelectrolyte) but, rather, a polyester. PLA is biodegradable, with similar characteristics to polypropylene (PP), polyethylene (PE) and polystyrene (PS). Notably, it can be produced from existing manufacturing equipment, which explains why PLA was quickly taken up by the plastic industries. There is a wide range of applications for PLA, such as the production of plastic bottles; films; and biodegradable medical devices (screws, pins, rods, and plates, with the expected capability to biodegrade within six to 12 months).

In order to mitigate membrane solid waste discharged to the environment, Szekely et al. [21] introduced another encouraging option to replace petroleum-based nonwoven backing materials (polypropylene or polyethylene terephthalate) commonly employed for flat sheet membranes. The work employed PLA and bamboo fibers to fabricate a porous membrane that can be used as the membrane backing (support) [21].

Bamboo fiber was chosen because of its abundance and low cost for production. The bamboo fiber contains a large amount of hydroxyl and ether groups, while the PLA is rich in hydroxyl groups and ester [21]. Thus, the hydrogen bonding and adhesion between the components can be facilitated. The bamboo fiber was first processed into dry particles in size of 50–150 μm [21], maintaining the original compositions composed of cellulose, hemicellulose, lignin, ash, and other extractives.

The mixture of bamboo fiber/PLA was dissolved in dimethyl carbonate (DMC), cast onto a glass substrate using a film applicator, and dried in a vacuum oven for 24 h at 30 °C, resulting in a suitable membrane support material similar to other nonwoven backings [21]. For testing the fabricated backing material, a dope solution of PBI in DMAc was prepared to be cast on the bamboo support; then, the filtration performance was tested. Various solvents were used for confirming the chemical stability of the support, such as DMF, NMP, DMSO, DMAc, and acetone, etc., and it was found to be stable in most of them for over six months. [21]. The research result clearly indicated that the bamboo-PLA membrane support could open a new door towards sustainable fabrication with green and biodegradable backing materials [21].

Recently, Lin Fan et al. [22] reported an interesting approach to fabricate thin film composite nanofiltration (TFC-NF) membranes via interfacial polymerization (IP) only in aqueous environments. Tannic acid and iron (III) chloride (FeCl_3_) dissolved in water acted as two reactive monomers. As a result, a stable coating layer containing metal-polyphenol was formed on a polyethersulfone (PES) porous support via the coordination reaction between tannic acid (TA) and iron ions (Fe^3+^). This green coating materials for NF membranes can be employed without any toxic reagents nor solvents. They also assessed how the reaction time, concentration of TA, and Fe^3+^ ions affect the permeation and separation properties of the formed membranes. Additionally, the membrane stability after immersion into different pH media and a NaClO solution was studied. Although the filtration experiment was conducted only for 24 h, the water flux and dye rejection ratio of the membranes remained stable. A conventional IP typically employs an organic layer. In comparison, this work showed that a thin surface layer can be formed at the surface in an aqueous environment. The work showed that TA-Iron(III)/PES composite NF membranes can be developed with green materials and reagents with excellent antioxidant properties after long-term immersion in a NaClO solution.

## 3. Sustainability for Employed Solvents

In pharmaceutical industries where solvents contribute a considerable mass intensity of products (up to 56%), a significant amount of research has been carried out to identify green solvents [37]. The undesired and preferred solvents from their perspectives are summarized in Figure 5. The key goal is to design and apply a greener synthesis route to avoid environmental impacts. Efforts should be given to move from the left side of the list in Figure 5 to the right, and chemical processes will gradually become greener and more sustainable. The solvents in the list were also referred from the work conducted by Alder et al. [38]; they updated and expanded GSK’s solvent sustainability guidelines conducted previously [39]; more solvents were supplemented, and the way in which the health, environment, safety, and waste categories were combined was adjusted.

As more applications develop using membrane technology, the environmental impacts caused by membrane preparation become more significant [15]. In membrane preparation, solvents play a crucial role, and the properties of a solvent and its interaction with the polymer affects the membrane morphology and, thus, performance [41]. Hence, identifying a green solvent that can dissolve the polymer of interest is only a prerequisite, as the resulting membrane must exhibit a competitive performance to be adopted by the membrane industry. Furthermore, environmental regulations now restrict the use of toxic solvents for membrane production. Notably, the European Registration, Evaluation, Authorization, and Restriction of Chemicals (REACH) has classified DMF, DMAc, and NMP as substances with very high concerns, and the use of these solvents will be restricted after May 2020 [15]. Recent reports on green solvent alternatives for membrane fabrication are summarized in Table 2.

### 3.1. Water

The solvents widely employed in the chemical industries are often organic chemicals from petroleum products with risks to health and the environment. Without a doubt, replacing organic solvents with water provides enormous benefits. A research work by Hanafia et al. [30] reported the preparation of a porous membrane from a water-soluble biopolymer, hydroxypropyl cellulose (HPC). HPC was first dissolved in water in a concentration of 20, and a 0.5% wt glutaraldehyde crosslinker was added in a dope solution after degassing to fix the membrane morphology and to prevent resolubilization. Just before casting the membrane, 1% wt HCl as a catalyst was added to initiate the crosslinking reaction.

The phase inversion was induced by elevating the temperature above the lower critical solution temperature (LCST) of the prepared solution. The membrane exhibited porous characteristics when analyzed with SEM. The membrane stability in various organic solvents (DMSO, tetrahydrofuran (THF), methanol, and chloroform) were confirmed showing the effectiveness of chemical crosslinking. This work has opened a new paradigm to fabricate polymer membranes using water as the solvent, following the green chemistry principles [30].

### 3.2. γ-Valerolactone (GVL)

There are some classical solvents that could potentially be considered as green solvents. Mainly, GVL is a solvent with a low melting and high flash point; it has a characteristic herbal odor, and recently, it has been used in flavor, perfume, and food industries [42]. GVL exhibits advantages such as a stable liquid form at ambient conditions, very low toxicity (LD50 oral rats = 8800 mg/kg) [43], high miscibility with water, and good stability in neutral media [23,44]. Importantly, GVL can be produced from cellulose-based biomasses [43].

With the objective of exploring bio-based green solvents in membrane preparation, Vankelecom et al. [23] prepared porous membranes via NIPS with common polymers (PI, PES, polysulfone (PSU), cellulose acetate (CA), and cellulose triacetate (CTA)) using GVL and a set of glycerol derivatives [23]. The employed bio-based cosolvents were α,α′-diglycerol, glycerol, monoacetin, glycerol carbonate, and diacetin. The work reported successfully the preparation of microfiltration (MF) and ultrafiltration (UF) membranes and, potentially, NF by increasing the dope polymer concentration.

### 3.3. Methyl Lactate and Ethyl Lactate

Figoli et al. (2014) [5] thoroughly reviewed the potential green solvents. Among the candidates, the lactate family is a group of nontoxic and biodegradable solvents with outstanding characteristics [5] that can replace toxic and environmental-unfriendly ones. Lactate esters can be obtained from bio-derived carbohydrate compounds, and the recent development of purification processes lowered the price for lactate products remarkably [24]. Additionally, it has been employed as an intermediate for polymers and chemicals such as basic dyes and herbicide [24].

Methyl lactate has a high miscibility with water, allowing it to function as a versatile solvent for membrane preparation via NIPS. Vankelecom et al. [24] made use of this solvent’s outstanding green characteristics to fabricate asymmetric cellulose acetate (CA) NF membranes via NIPS. The CA membrane showed a NF performance with 99.5% rejection for rose bengal (1017 MW Da) dye with 2.4-L/(m^2^·h·bar) permeance in water [24]. Meanwhile, the authors investigated the effects of cosolvent (2-methyl THF) up to 30–50% wt in dope solution, which proportionally made the membrane looser [24].

Ethyl lactate is another important monobasic ester found in small amounts in foods such as chicken, wine, and some fruits [45]. Ethyl lactate can be obtained by the esterification of ethanol and lactic acid [46]. This solvent has attracted considerable attention recently, as it satisfies at least eight of the “twelve principles of green chemistry” [45], with desirable properties of low viscosity, high boiling point, low vapor pressure, and low surface tension [45].

### 3.4. Cyrene

One of the emerging green solvents for replacing polar aprotic solvents (e.g., NMP, DMAc, and DMF) is the bio-based dihydrolevoglucosenone (Cyrene) [47]. Cyrene currently is a commercially available and nontoxic solvent obtained from renewable waste and a nonfood cellulosic source [25]. According to the work carried out by Sherwood et al. [47], Cyrene can be produced via two simple steps from the biomass, which can cut down on environmental footprints. The computer-aided Kamlet-Abboud-Taft (KAT) solvatochromic parameters indicate that Cyrene is aprotic, with an equivalent π* to those of strongly dipolar aprotic solvents but with a slightly lower β value (hydrogen bond-accepting ability) [47]. Interestingly, the solvent properties of Cyrene are very comparable to NMP without nitrogen or sulfur heteroatoms, which can result in NO_x_ and SO_x_ emissions upon disposal, reducing the environmental impact [47].

The application of Cyrene in membrane preparation was reported by Figoli et al. [48] for preparing PES and PVDF membranes by phase inversion, where the membranes were fabricated via simultaneous VIPS and NIPS methods [48]. In this work, the exposure time to atmospheric relative humidity was changed between 0 and 5 min to gain membranes with different pore sizes and permeability. Pure water permeability tests gave promising data for membranes fabricated via NIPS (∼23,400 L/m^2^/h/bar) and VIPS-NIPS (~18,700 L/m^2^/h/bar) [48]. The obtained results indicated the possibility to use Cyrene as a solvent for two of the most used polymers in the membrane industry. Additionally, the experimental results were discussed in connection to the viscosity of the dope solution, membrane morphology, ternary phase diagram, porosity, thickness, and contact angle [48].

Cyrene has also been used by other researchers to prepare PES (with polyvinylpyrrolidone (PVP) used as an additive) flat sheet membranes for water filtration application by NIPS technique [25]. An extensive comparison was made between membranes prepared using Cyrene and NMP. Membranes prepared with bio-based Cyrene were more sustainable, with less polymer losses, tunable pore sizes, and contact angles [25]. The authors claimed that membranes prepared with Cyrene exhibited higher total porosity and larger pore sizes in comparison with those fabricated using NMP [25]. By changing the temperature of the dope solution, which affects the viscosity of the dope solution, the permeability of membranes prepared with Cyrene was easily controlled in the range between NF/RO to MF applications [25].

Further testing of this solvent and other derivatives of Cyrene has been conducted to get more understanding about the replacing capacity of these bio-based solvents for NMP and similar solvents in filtration applications [47].

### 3.5. Polarclean

More recently, another promising eco-friendly alternative to replace polar aprotic solvents (e.g., NMP, DMAc, and DMF), Rhodiasolv^®^ Polarclean (hereafter referred to as Polarclean), was first suggested by Lee and Drioli et al. [49] for membrane preparation. The IUPAC name of the solvent is methyl-5-(dimethylamino)-2-methyl-5-oxopentanoate, and it is featured by a high solvent power and low volatility [49]. Polarclean has been employed as a solvent for agrochemicals and as a crystal growth inhibitor in cold solutions. Furthermore, its solubility in water is very high at 490 g/dm^−3^ at 24 °C [49].

Polarclean was first applied as an environmentally benign TIPS solvent to prepare PVDF membranes [10]. A comprehensive investigation was conducted to understand the underlying phenomena in the membrane formation kinetics during the TIPS process, and it was shown that the NIPS effect cannot be ignored at the dope-nonsolvent interface upon immersion into the water bath [10]. This phenomenon is now referred to as N-TIPS.

Combining NIPS and TIPS (e.g., N-TIPS) is a versatile technique to prepare membranes with narrow pore size distributions. This, however, can lower the membrane performance due to the formation of a dense skin layer by mass exchange at a solvent-nonsolvent interface [26]. Another simple and effective solution was suggested by Jung and et al. [26] by employing a triple spinneret to apply in a transient coating layer to eliminate the NIPS effect. The authors investigated the effects of different solvents in forming macro-porous hollow-fiber membranes using Polarclean as the dope solvent [26].

The Polarclean solvent was then employed as a NIPS solvent to prepare PI, CA, PES, and PSU membranes by Wang et al. [14]. The work showed that MF, UF, NF, and gas separation membranes can be fabricated, showing the versatility of this environmentally benign Polarclean solvent.

In another work carried out by Marino et al. [50], porous PES membranes were successfully fabricated with high pure water permeability. The prepared membranes were fabricated via the combined NIPS and VIPS technique using PolarClean as the solvent. Use of common additives such as PVP and poly(ethylene glycol) (PEG) affected the membrane morphology [50].

### 3.6. Triethyl-Phosphate (TEP)

TEP is a widely used solvent in the industrial scale as an intermediate for insecticide products, as a catalyst for anhydride synthesis, and as an agent for manufacturing rubbers and plastics. TEP appears as a colorless, corrosive liquid with a mildly pleasant odor. Although TEP is considered harmful if swallowed (Regulation No. 1272/2008 by the EU), this substance contains no components associated with bioaccumulation [27].

There are many works that employed TEP for membrane preparation [27,51,52]. It can be applied as both a NIPS and TIPS solvent. Notably, Sufyan Fadhil et al. [27] reported research on fabricating poly(vinylidene fluoride-hexafluoropropylene) (PVDF-HFP) flat sheet membranes using a TEP solvent prepared for direct contact membrane distillation (DCMD) [27]. The authors employed a phase inversion technique with various coagulation bath conditions to optimize the membrane performance for DCMD.

### 3.7. Ionic Liquid (IL) and Deep Eutectic Solvents (DES)

An ionic liquid (IL) is a salt that exists in the state of a liquid, composed of ions and short-lived ion pairs. In the context of green solvents, ILs have two features that can be considered green, depending on their application. First, ILs have negligible vapor pressure (low volatility) without concerns over volatile organic compounds. ILs have been applied in membrane technology rather often recently. Cellulose hollow fibers were fabricated by spinning with a dope solution comprising 1-ethyl-3-methylimidazolium acetate ((EMIM)(Ac)), 1-ethyl-3-ethylimidazolium diethyl phosphate ((EMIM)(DEP)), or 1,3-dimethylimidazolium chloride ((EMIM)(Cl)) [28]. Secondly, ILs allow membrane preparation by a single-step procedure, which means cellulose (nonsoluble in most solvents) can be directly dissolved in the target solvent. A similar work by Sukma et al. [29] reported the preparation of NF cellulose membranes via phase inversion using 1-ethyl-3-methylimidazolium acetate ((EMIM)OAc) as the solvent and acetone as the volatile cosolvent [29].

Abdellah et al. [51] successfully fabricated a polyester TFC OSN membrane on a cellulose substrate that was prepared using IL as the solvent. In addition, the active layer was composed of quercetin (a polyphenol normally found in blueberries, grapefruit, and black tea) crosslinked with terephthaloyl chloride.

In comparison to the currently employed organic solvents, ILs undeniably stand as a potential alternative with excellent thermal stability, low chemical reactivity, and negligible vapor pressure without flammability. However, in spite of such advantages, some concerns remain over the green-ness of the ILs, including unsustainable synthesis protocols [44], high cost, and uncertain biodegradability in many types of ILs [44]. Hence, further research to carefully assess the sustainability of ILs must be performed.

On the other hand, deep eutectic solvents (DES) are considered as a new generation of ILs and have emerged as a biodegradable and renewable green solvent. DES are available at a much lower price [53], and they can be produced without generating excessive waste. This solvent group has brought comparable physicochemical properties to those of ILs. In comparison to ILs, DES can be easily prepared from the available starting materials with universal dissolution abilities, extensive tenability, and ease of synthesis [54].

Though DES has not been fully employed as a solvent for membrane fabrication yet, some researchers have been uncovering DES as flux boosting and surface cleaning agents for TFC polyamide membranes [54]. A considerable flux increase was obtained after treating polyamide TFC by DES at low temperatures without a noticeable change in the rejection performance. The authors asserted such an improvement is due to the improved wettability on the surface of DES-treated membranes, which enhanced the smoothness of a polyamide TFC membrane in the process of DES treatment [54]. The work reused the DES for activating the surface of polyamide TFC membranes for six consecutive cycles.

In another work carried out by Esmaeili [53], DES (containing choline chloride and lactic acid in a molar ratio of 1:9) was used to extract lignin. The DES-lignin 0–1% wt mixture was employed as a hydrophilic adhesion promoter to fabricate antioxidant PES membranes by the phase inversion technique [53]. In this work, NMP was used as the main solvent. The membrane properties were investigated in terms of effect of DES-lignin on the antioxidant activity and membrane performance [53]. Through the obtained results, DES-lignin-modified membranes showed improved hydrophilicity, antioxidant activity, and high pure water permeability, with less negative surface charge in comparison to a pristine PES membrane, while rejection remained almost constant [53]. Although the work used NMP as the dope solvent, other green solvents could be employed in further studies in the future.

### 3.8. Other Green Solvents (Non-OSN Membranes)

Since the TIPS technique proceeds at higher temperatures (better polymer solubility), more possibilities are available to identify green solvents. Notably, one of the recently emerging green solvents is acetyl tributyl citrate (ATBC) without adverse health hazards. ATBC has been reported and widely used as a plasticizer for pharmaceutical coatings and food packaging [11]. ATBC is not miscible with water, and it is considered to be much more environmentally benign in comparison with phthalate-based TIPS solvents [11]. ATBC was employed to fabricate a PVDF membrane with competitive performance. Up to 50% wt of the polymer concentration was used in order to maximize the mechanical strength of the membranes, and the prepared membrane was then stretched to enhance the porosity and permeability [11].

The idea of using universal crystallizable diluent to fabricate a porous membrane by TIPS has been proposed in some reports, and one of the key points herein was seeking for a proper diluent based on the crystallinity and high melting point of polar polymers [31]. Fan et al. [31] employed dimethyl sulfone (DMSO_2_) as a universal crystallizable TIPS diluent for polar polymer membranes of PVDF, PAN, and CA, because the solvent’s solubility parameter is close to those of the polymers [31]. The tensile strength and elongation of the prepared membranes could be improved if the polymer concentration or cooling rate was increased. Notably, recrystallization and sublimation can be applied to recover and reuse DMSO_2_. This work can be considered as a green preparation method for microporous polymer membranes via TIPS [31].

Chaouachi et al. [55] reported the use of butyl acetate as a greener solvent to fabricate membranes from recycled low-density polyethylene (LDPE) via the TIPS method [55]. LDPE is a semi-crystalline polymer, soluble only at elevated temperatures in toxic organic solvents such as phthalate-based compounds [55]. The authors made a comparison using xylene as a solvent to assess the difference, and the results indicated that the membrane prepared using butyl acetate showed desirable properties in terms of thickness (~0.363 mm), pore size (~0.14 μm), and contact angle (~120°). These results were encouraging, because the polymer was valorized from recycled sources [55].

## 4. Sustainability for Membrane Fabrication Procedure

### 4.1. Minimizing the Number of Fabrication Steps and Materials

Similar to reverse osmosis technology, the trend in OSN membranes is shifting from asymmetric to TFC membranes that guarantee better performance. Recently, a TFC membrane with 10-nm selective layer thickness with ultrafast solvent permeance was reported [56]. In the TFC membrane, the selective layer mostly governs the separation performance, and the support layer provides the necessary mechanical support and handleability [2]. The performance of TFC membranes can be affected significantly by the support layer, and thus, optimization of the support layer morphology is also important. Particularly for OSN, the support layer also must be solvent-stable, limiting the number of options available.

Recently, Lee et al. [32] employed a commercially available PE battery separator as a porous support to prepare TFC OSN membranes with impressive performances [32]. A porous PE battery separator is a fascinating support material for OSN applications, as it exhibits a very high solvent resistance at an ambient temperature with outstanding mechanical strengths.

The fabrication procedure is illustrated in Figure 6. It can be seen that, in comparison to the conventional TFC-PI membrane process, the TFC-PE membrane can be prepared with just two steps: O_2_ plasma for hydrophilization (20 s), followed by interfacial polymerization (10~15 min). Superior performances were obtained across all ranges of solvents with TFC-PE membranes compared to the conventional TFC-PI membranes [32]. In comparison, fabrication of the TFC-PI membrane can take up to three to four days, starting from dissolution of the polymer to a lengthy crosslinking reaction period.

In another work by Livingston et al. [33], a robust TFC OSN membrane was fabricated using the spray-coating technique, with excellent stability in high temperatures and fast solvent permeability [33]. The details of the proposed method are schematized in Figure 7. It can be seen that fabrication of the conventional asymmetric OSN membranes can take up to one week and requires 0.11 kg of polymer per m^2^ of membrane. On the other hand, the proposed method only takes half a day and requires only 0.02 kg of polymer per m^2^ of membrane. The prepared membranes were tested up to 15 days in DMF solution at different temperatures [33].

This method has several advantages from the green perspective, such as a lower carbon footprint, lower consumption of solvent and polymers, and shorter time for the coagulation and washing steps [33].

### 4.2. Sustainable Post-Modification

OSN membranes must show solvent resistance, and the crosslinking technique is the most widely employed method to improve the polymer solvent stability. Generally, a crosslinking step employs toxic diamines, dihalide, diols, and other reactive chemicals in organic solvents at high temperatures [34]. The conditions required for crosslinking are very harsh and certainly not green.

Szekely et al. [34] proposed a unique method to improve the polymer chemical stability without crosslinking the backbone. The work employed the concept of interpenetrating polymer networks (IPN) using dopamine monomers to fabricate PBI/polydopamine (PDA) OSN membranes with unprecedented thermal and chemical stability. Surprisingly, the PBI backbone was not chemically crosslinked, yet exhibited a solvent stability in polar aprotic solvents (DMF, NMP, and DMAc), even at temperatures above 100 °C. A close examination revealed that the dopamine monomer formed another intrachain network within the PBI backbone, forming a strong H-bond with the PBI imidazole groups. Hence, PBI and PDA were physically crosslinked in a form of semi-IPN. A density functional theory (DFT) analysis revealed that the H-bond between polydopamine (PDA) and PBI’s imidazole groups were stronger than the solvation energy by aprotic solvents, supporting the observed solvent stability.

The notable part is the sustainability achieved for the fabrication of such membranes. The key step to fabricate this membrane, as shown in Figure 8, was the in-situ polymerization of the dopamine within a PBI membrane. A dope solution comprising PBI and dopamine monomers was prepared; then, it was cast and phase-inverted. The dopamine monomer, during this stage, plays a role as an additive. The polymerization of the remaining dopamine monomer was carried out in the aqueous media of a mixed NaIO_4_ and Tris buffer solution, yielding a semi-IPN membrane with PDA and PBI. The stability and performance of the fabricated membranes with different polymerization times were tested at fixed pressures and temperatures. Compared to the pristine PBI membranes, the PDA/PBI-IPN membranes showed a robust stability in harsh aprotic solvents with good NF performances.

In another work of Szekely et al. [35], the performance of the OSN membrane was enhanced by a coating with bio-phenol materials. From the economical perspective, plant-originated bio-phenols can be low-cost alternatives. The work employed tannic acid (originated from cork oak), vanillyl alcohol (from vanilla pods), eugenol (extracted from cloves), morin (guava leaves), and quercetin (common flavonol amongst plants). First, a PBI membrane was cast from a 19% to 24% wt polymer dope solution, which was homogeneously dissolved, on a polypropylene nonwoven support. The pristine membrane was then immersed in a solution containing the aforementioned bio-phenols for bio-coating. This step was followed by immersion into a NaIO_4_ solution for complete coating. The solvent resistance of the prepared membranes were investigated by soaking them in eight common organic solvents and four green solvents [35]. The membrane performance indicated that the bio-phenol coating resulted in a denser membrane with a lower permeance (22–92%) and higher rejection (12–79%). Importantly, the work also demonstrated that the membrane swelling could be cut down by up to 80% after coating.

## 5. Conclusions

Improving the sustainability of membrane fabrication is an important and pressing issue in the field of membrane technology, requiring more efforts and attention. Perhaps the most pressing matter is the fact that the use of common polar aprotic solvents (DMF, NMP, and DMAc) will be restricted in May 2020. Although the application range of membrane technology is very wide, we mostly focused on the sustainability of OSN, particularly of polymeric membranes. As compiled in this review, many of the works focused on identifying greener alternatives. However, recently, many greener fabrication protocols have also been reported to minimize fabrication mass intensity.

With increasingly strict regulations on environmental protection, research interest should be paid to not only the greener fabrication process of membranes but, also, the whole life cycle of membrane fabrication. It is necessary to carry out comprehensive research by following the “green” concept for the solvent-solute membrane system. Moreover, sustainable fabrication could be more meaningful if experimental and optimal conditions were provided, setting the basis for a sustainable analysis. With updated applications and trends in the fabrication of polymeric membranes, hopefully, encouraging movements will be further carried out in the path of achieving sustainability.

## Figures and Tables

**Figure 1 membranes-11-00019-f001:**
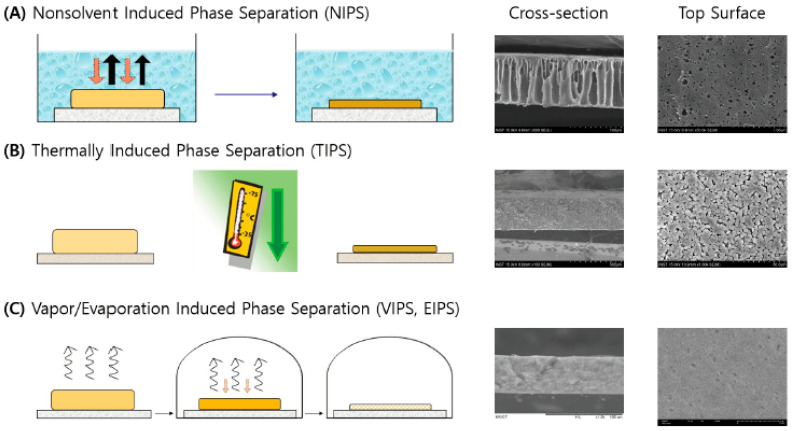
Illustration of the phase inversion technique and respective membrane morphology for (**A**) NIPS (reprinted with permission from) [10], (**B**) TIPS (reprinted with permission from) [11], and (**C**) VIPS and EIPS (reprinted with permission from) [12].

**Figure 2 membranes-11-00019-f002:**
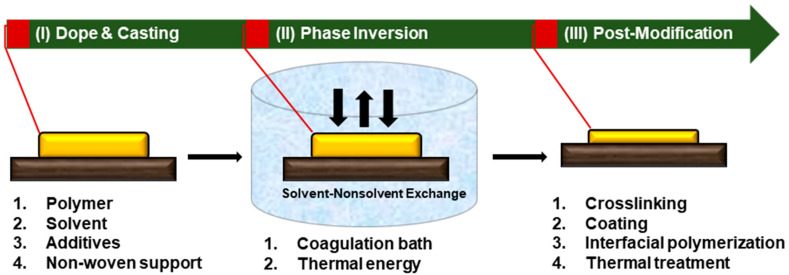
Important parameters and components involved in each step of the phase inversion method.

**Figure 3 membranes-11-00019-f003:**
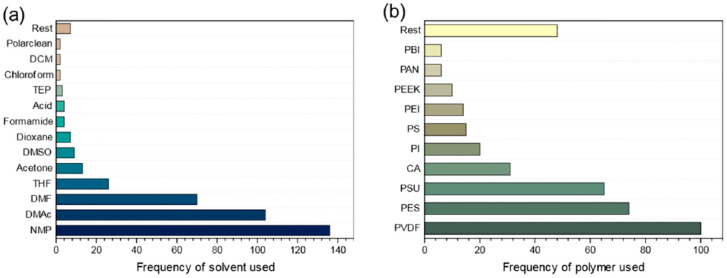
Literature survey result of the (**a**) solvents and (**b**) polymers employed in NIPS membrane fabrication over the past five years (reprinted with permission from [14]).

**Figure 4 membranes-11-00019-f004:**
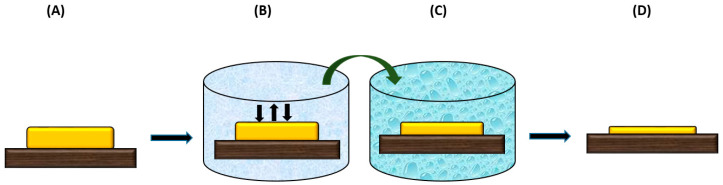
Graphical illustration of the RIPS technique (**A**) casting a thin film with polymer solution, (**B**) immersing the cast film in an aqueous calcium chloride bath, (**C**) immersing the film in a nonsolvent bath, and (**D**) the final membrane.

**Figure 5 membranes-11-00019-f005:**
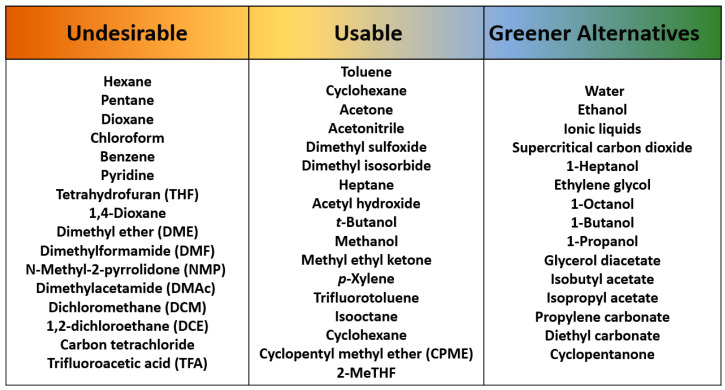
Recommendations for selecting solvents in industrial processes with reference to GSK’s solvent selection guide; each level lists undesirable, usable, and preferred solvents, respectively [38,40].

**Figure 6 membranes-11-00019-f006:**
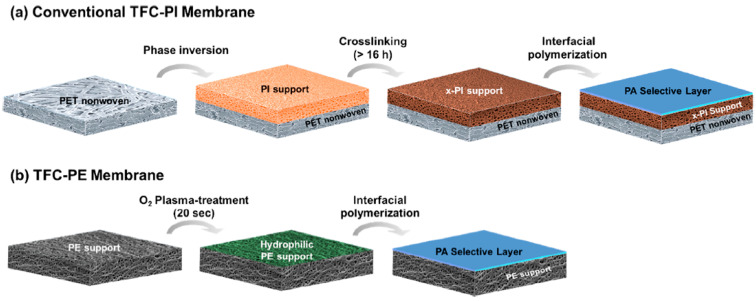
Comparison of fabrication protocols for (**a**) the TFC-PI membrane prepared on crosslinked polyimide (PI) support [57,58] and (**b**) proposed TFC-PE membrane on plasma-treated polyethylene (PE) support [32] (reprinted with permission from [32]).

**Figure 7 membranes-11-00019-f007:**
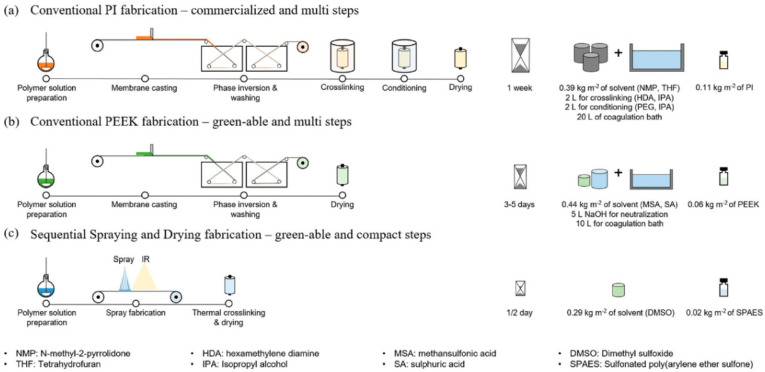
Schematic diagrams for the conventional fabrication methods to make (**a**) commercialized PI organic solvent nanofiltration (OSN) membranes, (**b**) poly(ether ether ketone) (PEEK) OSN membranes, and (**c**) proposed sequential spraying and drying fabrication methods with compact steps [33] (reprinted with permission from [33]).

**Figure 8 membranes-11-00019-f008:**
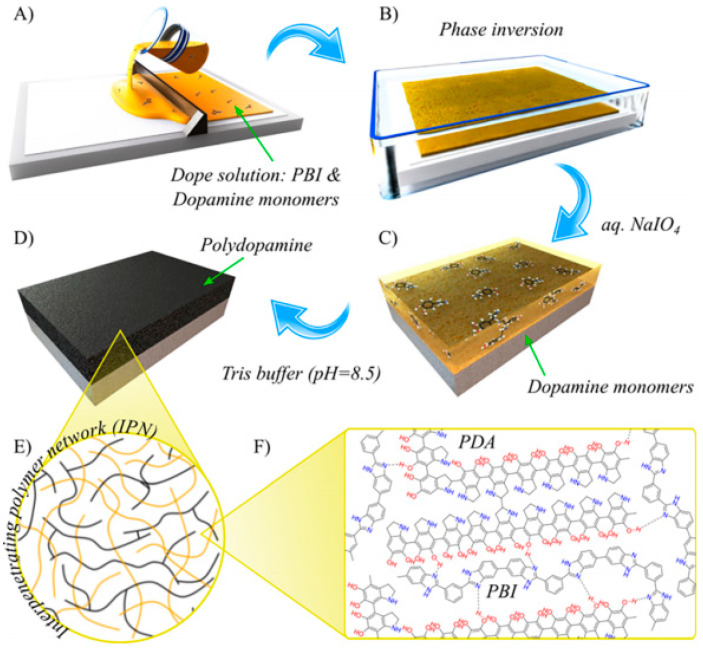
Process of preparing a PBI nanofiltration (NF) membrane applying the interpenetrating polymer network (IPN): (**A**) casting the membrane from the dope solution containing PBI and polydopamine (PDA) and (**B**) immersing the membrane in a coagulation bath; membrane was immersed in (**C**) NAIO_4_ and (**D**) Tris buffer solution, subsequently. (**E**,**F**) illustrate the IPN formed between PBI and PDA (reprinted with permission from [34]).

**Table 1 membranes-11-00019-t001:** Performance of membranes fabricated with sustainable materials and solvents.

No.	Selective Membrane Material	Support Material	Solvent for Fabrication	Testing Solvent	Marker (MW)	Permeance (L/m^2^·h·bar)	Highest Rejection (%)	Ref.
1.	NaAlg	PAN	Water	Methanol	B12 (1355 g·mol^−1^)	1.27 ± 0.2	98 ± 2	[18]
2.	NaAlg	Crosslinked PAN	Water	DMF	B12 (1355 g·mol^−1^)	0.21 ± 0.1	98 ± 1	[18]
3.	NaAlg	Cellulose	Water	Methanol	B12 (1355 g·mol^−1^)	0.38 ± 0.1	95 ± 1	[18]
4.	NaAlg	Alumina support	Water	Methanol	B12 (1355 g·mol^−1^)	1.6 ± 0.1	90± 2	[18]
5.	NaAlg	Alumina support	Water	DMF	B12 (1355 g·mol^−1^)	0.25 ± 0.02	70 ± 1	[18]
6.	NaAlg	Alumina support	Water	DMSO	B12 (1355 g·mol^−1^)	0.15 ± 0.05	76 ± 2	[18]
7.	NaAlg	Alumina support	Water	NMP	B12 (1355 g·mol^−1^)	0.11 ± 0.03	80 ± 3	[18]
8.	PBI	Bamboo fiber/PLA	DMC, DMAc	Water		1068 ± 32		[21]
9.	CTA	Polypropylene nonwoven	GVL	Water	Rose Bengal (1017 g·mol^−1^)		>90	[23]
10	CA	Polypropylene nonwoven	Methyl lactate	Water	Rose Bengal (1017 g·mol^−1^)		>90	[23]
11.	CA	Polypropylene nonwoven	GVL	Water	Rose Bengal (1017 g·mol^−1^)		>90	[23]
12.	CA	Polypropylene nonwoven	Methyl lactate 2-methyl THF	Water	Rose Bengal (1017 g·mol^−1^)	2.4	99.5	[24]
13.	PES	Polyester nonwoven	Cyrene	Water		2542.7		[25]
14.	PES	Polyester nonwoven	Cyrene	Water		898.4		[25]
15.	PVDF		PolarClean	Water		3000		[10,24]
16.	PVDF		PolarClean	Water	Polystyrene		99.99	[26]
17.	PVDF-HFP		TEP	Distilled water	NaCl	16.1	99.3	[27]
18.	Cellulose		((EMIM)(DEP))	Ethanol	Congo Red (696 g·mol^−1^)	19 ± 1	>90	[28]
19.	Cellulose		((EMIM)(DEP))	Water	Congo Red (696 g·mol^−1^)	48 ± 3	>99	[28]
20.	Cellulose		(EMIM)OAc	Ethanol	Bromothymol Blue (624.4 g·mol^−1^)	0.3	94	[29]
21.	Cellulose		(EMIM)OAc/Acetone	Ethanol	Bromothymol Blue (624.4 g·mol^−1^)	8.4	69.8	[29]
22.	TA/Fe^3+^	PES	Water	Water	Orange GII (452.4 g·mol^−1^)	45.6	94.8	[22]
23.	TA/Fe^3+^	PES	Water	Water	Orange GII (452.4 g·mol^−1^)	34.3	95.5	[22]
24.	HPC		Water	Water		3 ± 0.2 − 38 ± 5		[30]
25.	PVDF		ATBC	Water		538		[11]
26.	PVDF		DMSO_2_	Water		1491		[31]
27.	PA	PE battery separator	Hexane, Water	Acetone	Styrene oligomer (~1000 g·mol^−1^)	~20	>99	[32]
28.	PAES	Porous substrate(PE separator, TR-NFM, and PET nonwoven)	DMSO	DMF	Styrene oligomer (1595 g·mol^−1^)	0.37 ± 0.018	>99	[33]
29.	PAES, PES-TA	PAES membrane with PES-TA group	PEGDGE 10% wt aqueous solutions, PEI% wt solution	Water	Methyl violet (407.979 g·mol^−1^)	15.5	99.8	[6]
30.	PBI/PDA	Polypropylene	DMAc, Water	DMF	Polystyrene (610 g·mol^−1^)	12	~100	[34]
31.	Bio-phenol coated PI	PI membrane	Water:EtOH	Acetone	Polystyrene (390–1550 g·mol^−1^)	1–10	5~100	[35]
32.	PI	Crosslinked PI membrane	DMSO	Methanol	Sunset Yellow (452 g·mol^−1^)	11	93	[36]

**Table 2 membranes-11-00019-t002:** Potential green solvent candidates.

No.	Solvent	Chemical Structure	Soluble Polymer	Sustainable Characteristics	Ref.
1.	Water	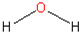	HPC	An inorganic, transparent, tasteless, odorless and nearly colorless chemical substance with low risk of hazard, low cost and high availability	[30]
2.	γ-Valerolactone (GVL)	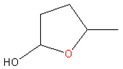	PI, PES, CA, CTA, PSU	Colorless liquid, bio-based green solvent, low toxicity and miscible with water, stability under neutral media	[23]
3.	Methyl lactate	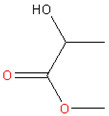	CA, cellulose derivatives	Colorless and clear substance with a peculiar odor, high boiling point and slow volatility rate with biodegradability, water miscibility, noncorrosive, noncarcinogenic, and non-ozone-depleting features	[23,24]
4.	Ethyl lactate	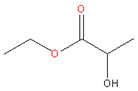	CA	Colorless, sweet smell and clear substance, low toxicity, and agreeable odor, water miscibility, noncorrosive, noncarcinogenic, and non-ozone-depleting features	[45]
5.	Cyrene	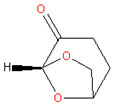	PES, PVDF	Bio-based origin, clear colorless to light yellow liquid, mild, smoky ketone-like odor, with a comparatively high dynamic viscosity, comparable solvent properties to NMP without nitrogen or sulfur heteroatoms	[47,48]
6.	Rhodiasolv^®^ Polarclean	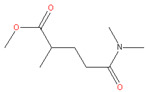	PI, CA, PVDF	Clear, colorless to yellow liquid with slight odor, solubility in water, high solvency capability, eco-friendly sustainable solvent	[10,24,26]
7.	Triethyl phosphate (TEP)	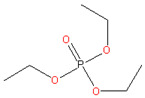	CA, PVDF	Colorless, corrosive liquid, combustible, slowly dissolves in water and sinks in water, no components supposed of persistence, bio-accumulation, and toxicity or high persistence in the environment	[27]
8.	Ionic Liquids (ILs)	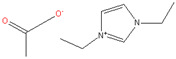 Ex) 1-ethyl-3-ethylimidazolium acetate	PAN, Cellulose, PES, CA	Low melting point, high thermal stability, low viscosity, low chemical reactivity, and negligible vapor pressure without flammability	[28,29]

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
