# Peer review of "Sustainable Fabrication of Organic Solvent Nanofiltration Membranes"

_membranes, 2020, doi:10.3390/membranes11010019_

Round 1
Reviewer 1 Report
This manuscript compiles recent progress and advances for sustainable fabrication in the field of polymeric OSN membranes. The manuscript was well organized, and provided many information. The work can be accepted for publication in Membrane after a minor revision
- In Figure 4, why toluene, cyclohexane, and acetone are in the usable list, while they are quite similar to benzene, hexane, heptane in the undesirable list?
- For organic solvent nanofiltration, the most important thing is the solvent resistance. So that the authors are recommended to present the long-term solvent resistance performance of the membranes fabricated with sustainable materials and solvents in Table 1, and also in the context. Also, the comparison of the solvent resistance using sustainable materials and solvents with the state-of-the-art literature work using traditional solvents is suggested.
- Line 154~165, the authors illustrated a work referenced from [21], while it is not OSN membrane, nor the work has some relation with OSN membrane. The authors are suggested to relate it to their main idea of OSN membranes.
- Section 4.1, for minimizing number of fabrication steps and materials, only the support layer was discussed. The skin layer is very important for the TFC membranes. The authors should add reference on the important influence of the skin layer in membrane sustainable fabrication.
Author Response
This manuscript compiles recent progress and advances for sustainable fabrication in the field of polymeric OSN membranes. The manuscript was well organized, and provided many information. The work can be accepted for publication in Membrane after a minor revision
- In Figure 4, why toluene, cyclohexane, and acetone are in the usable list, while they are quite similar to benzene, hexane, heptane in the undesirable list?
Answer: We thank the reviewer for the concern. According to the GSK Solvent Selection Guide (Green Chem., 2016, 18, 3879), toluene, cyclohexane, acetone and heptane also are listed as solvents with “some issues” while benzene, hexane, tetrahydrofuran (THF), and dimethylformamide (DMF) are allocated to the group of solvents with major issues. These solvents are marked with environment, health, and safety regulatory alert. In the supporting table under the Guide, cyclohexane, acetone and heptane were presented without legislation flag and toluene was tagged with “no current restrictions but future regulatory restrictions may apply”.
- For organic solvent nanofiltration, the most important thing is the solvent resistance. So that the authors are recommended to present the long-term solvent resistance performance of the membranes fabricated with sustainable materials and solvents in Table 1, and also in the context. Also, the comparison of the solvent resistance using sustainable materials and solvents with the state-of-the-art literature work using traditional solvents is suggested.
Answer: We agree with the reviewer that long-term stability should be discussed in the manuscript. Unfortunately, not many works conducted nor reported long term solvent resistance data. As per suggestion, we have appended long-term stability of reported membranes wherever possible in the text. We also hope that in the future, works related to OSN will indicate their membrane stability in prolonged experiments. For comparing solvent resistance using sustainable materials and solvents, the materials were stable in toxic polar aprotic solvents (NMP, DMF, etc.).
- Line 154~165, the authors illustrated a work referenced from [21], while it is not OSN membrane, nor the work has some relation with OSN membrane. The authors are suggested to relate it to their main idea of OSN membranes.
Answer: The title of the mentioned reference is “Green coating by coordination of tannic acid and iron ions for antioxidant nanofiltration membranes”, where membranes were fabricated via interfacial polymerization method on PES supports. Although the membrane may not exhibit stability in strong solvents such as NMF, NMP, etc., we decided to include it as the green fabrication method was applied for nanofiltration application. Also, the authors reported good structural stability without disassembly and delamination in acidic or alkaline solution with stable dye rejection after stability experiment.
4. Section 4.1, for minimizing number of fabrication steps and materials, only the support layer was discussed. The skin layer is very important for the TFC membranes. The authors should add reference on the important influence of the skin layer in membrane sustainable fabrication.
Answer: This suggestion is very valuable to our work, we have now revised the manuscript as per comment. Actually, the skin layer was discussed in line 154~165 with efforts to improve sustainable fabrication for selective layer of TFC membrane, it was not allocated in section 4.1 because the work just focused on employing environmentally friendly material without minimizing fabrication steps mentioned, thus we decided to present skin layer and support layer in different parts according to our approach.
Reviewer 2 Report
This review is dealing with Organic solvent nanofiltration (OSN) as one of the key technologies to improve the sustainability of separation processes.
It is well written but minor revisions regarding the English essu have to be considered.
Author Response
Reviewer #2
This review is dealing with Organic solvent nanofiltration (OSN) as one of the key technologies to improve the sustainability of separation processes.
It is well written but minor revisions regarding the English essu have to be considered.
Answer: We appreciate the reviewer’s incentive comments. We have now revised English writing of the manuscript according to the reviewer’s suggestions.
Reviewer 3 Report
This paper reviews the sustainable fabrication of OSN membrane recently, including the polymer, solvent and modifier. It makes sense. However, there are still some problems in this article. Therefore, it needs major revision. The detailed comments are listed below.
- In introduction, as a traditional method to prepare the membrane, NIPS has been extensively reported. Please refer to the following articles: 1) ACS Applied Materials & Interfaces, 2020, 12, 580-590; 2) Chemical Engineering Journal, 2020, 385, 123993.
- The author should clarify the solvent resistance mechanism of the traditional OSN membranes to facilitate the reader’s better reading.
- It is better to show an outlook in depth at the conclusion. For example, what is the trend of the green fabrication methods of OSN in future?
- The language should be polished by native speekers.
Author Response
This paper reviews the sustainable fabrication of OSN membrane recently, including the polymer, solvent and modifier. It makes sense. However, there are still some problems in this article. Therefore, it needs major revision. The detailed comments are listed below.
- In introduction, as a traditional method to prepare the membrane, NIPS has been extensively reported. Please refer to the following articles: 1) ACS Applied Materials & Interfaces, 2020, 12, 580-590; 2) Chemical Engineering Journal, 2020, 385, 123993.
Answer: We appreciate the reviewer’s suggested references. We have appended the suggested references in the revised manuscript.
- The author should clarify the solvent resistance mechanism of the traditional OSN membranes to facilitate the reader’s better reading.
Answer: We thank the reviewer’s constructive comment. We have now revised the manuscript according to the suggestions. The supplemented and revised content is copied below
“With growing environmental concerns, the concept of sustainability has become more important, and membrane technology is being widely accepted by industries with practical applications. Organic solvent nanofiltration (OSN) is a membrane process with capabilities to discriminate nanometer-sized molecules in organic solvents [4]. Different from ultrafiltration and reverse osmosis, nanofiltration membranes employ typical pore size distribution (0.5-2 nm) and complex mechanism for specific separation of small molecular organics and salts [7] OSN has been recognized as a sustainable separation platform with low energy consumption [8], and also as a platform to achieve process intensification via reaction-separation convergence [9].
In order to perform separation in organic media, OSN membranes are required to repel harsh media such as aggressive solvents of DMF, NMP as well as solvents with oligonucleotide reaction (acetonitrile) [9], at the same time, the membrane must be stable over a wide range of pH and strong to organic acids and bases [9]. Hence, most research efforts have been dedicated to improve the chemical stability of polymeric membranes. However, modifications such as crosslinking employ toxic reagents and chemicals that generate considerable amount of organic wastes. Ironically, although OSN technology has developed to improve the process sustainability, the fabrication of OSN membrane itself has been far from sustainable [8].”
- It is better to show an outlook in depth at the conclusion. For example, what is the trend of the green fabrication methods of OSN in future?
Answer: Thank you for this valuable suggestion. We have revised the manuscript to include future trends in OSN membrane fabrication methods. The revised conclusion is copied below.
“Improving the sustainability of membrane fabrication is an important and pressing issue in the field of membrane technology, requiring more efforts and attention. Perhaps the most pressing matter is the fact that the use of common polar aprotic solvents (DMF, NMP, and DMAc) will be restricted from May 2020. Although the application range of membrane technology is very wide, we mostly focused on the sustainability of OSN, particularly of polymeric membranes. As compiled in this review, many of the works have been focused on identifying greener alternatives. However, recently, promising greener fabrication protocols have also been reported to minimize fabrication mass intensity.
With increasingly strict regulations on environmental protection, research interest should be paid on not only greener fabrication process of membrane but also the whole life cycle of membrane fabrication. It is necessary to carry out comprehensive research by following “green” concept for solvent-solute-membrane system. Moreover, the sustainable fabrication could be more meaningful if experimental and optimal conditions were provided, setting the basis for sustainable analysis. With updated applications and trends in fabrication of polymeric membrane, hopefully, encouraging movements will be further carried out in the path of achieving sustainability.”
- The language should be polished by native speakers.
Answer: We appreciate the reviewer’s suggestion. We have tried to improve the English language throughout manuscript.
Round 2
Reviewer 3 Report
This manuscript has been improved by the authors. It can be accepted for publication.